# Mechanotransduction in Skin Inflammation

**DOI:** 10.3390/cells11132026

**Published:** 2022-06-25

**Authors:** Maria S. Shutova, Wolf-Henning Boehncke

**Affiliations:** 1Department of Pathology and Immunology, University of Geneva, 1211 Geneva, Switzerland; wolf-henning.boehncke@hcuge.ch; 2Department of Dermatology, Geneva University Hospitals, 1211 Geneva, Switzerland

**Keywords:** keratinocyte, epidermis, psoriasis, atopic dermatitis, fibrosis, integrins, actin-myosin cytoskeleton, intermediate filaments, cytokines, stretch

## Abstract

In the process of mechanotransduction, the cells in the body perceive and interpret mechanical stimuli to maintain tissue homeostasis and respond to the environmental changes. Increasing evidence points towards dysregulated mechanotransduction as a pathologically relevant factor in human diseases, including inflammatory conditions. Skin is the organ that constantly undergoes considerable mechanical stresses, and the ability of mechanical factors to provoke inflammatory processes in the skin has long been known, with the Koebner phenomenon being an example. However, the molecular mechanisms and key factors linking mechanotransduction and cutaneous inflammation remain understudied. In this review, we outline the key players in the tissue’s mechanical homeostasis, the available data, and the gaps in our current understanding of their aberrant regulation in chronic cutaneous inflammation. We mainly focus on psoriasis as one of the most studied skin inflammatory diseases; we also discuss mechanotransduction in the context of skin fibrosis as a result of chronic inflammation. Even though the role of mechanotransduction in inflammation of the simple epithelia of internal organs is being actively studied, we conclude that the mechanoregulation in the stratified epidermis of the skin requires more attention in future translational research.

## 1. Introduction

Mechanotransduction is defined as the capability of cells to sense and transform mechanical cues into biochemical signals. The cells perceive the mechanics and composition of their environment primarily through elaborate adhesive cytoskeletal modules comprising the cell–cell and cell–matrix adhesions. The dynamic association of the transmembrane adhesion receptors with the cytoskeleton inside the cell orchestrates mechanosensitive pathways that transmit signals to the nucleus to fine-tune the gene expression, defining the cell phenotype and fate and maintaining the overall form and function of the tissue. Thus, the mechanical environment and its proper perception by the cell are critical for cell proliferation, motility, and differentiation [1,2].

To promote mechanical homeostasis in health, cells must use negative feedback mechanisms that sense changes within the extracellular environment and restore it back to normal. Diverse pathologies seem to result from a loss of negative feedback and a switch to positive feedback mechanisms involving the upregulation of certain signaling cascades. The altered tissue mechanics and/or cellular mechanosignaling lead to increased cell proliferation and motility that are characteristic for cancer as well as cardiovascular, neurodegenerative, fibrotic, and inflammatory disorders and abnormal wound healing [3,4].

Skin is the largest human organ and has constant intense contact with the environment via exposure to mechanical damage and trauma, temperature fluctuations, irradiation, chemicals, allergens, infections, etc. These numerous factors affect skin homeostasis, in part via the alteration of mechanosensitive pathways, and this can be implicated in skin inflammatory disorders.

Three layers can be distinguished in mammalian skin. The deepest layer is the subcutaneous layer, formed by adipocytes and connective tissue. It serves the purposes of insulation, fat storage, and protection of the internal body organs. The following layer above it is the dermis, which predominantly consists of an extracellular matrix (ECM) produced by the dermal fibroblasts. The dermal ECM contains high amounts of collagen and elastin, which provide mechanical stability and elasticity to the skin. The dermis also comprises blood vessels, mechano- and thermoreceptors, hair follicles, sweat glands, and sebaceous glands, and is populated by immune cells such as mast cells, macrophages, and lymphocytes. The top layer of the skin is the epidermis, a stratified squamous epithelium tightly attached to the dermis by the basement membrane (BM) (Figure 1A). About 90 percent of the epidermal cells are keratinocytes. The epidermal basal layer (stratum basale) is connected to the BM and contains proliferating keratinocytes that supply the cells for the upper layers and are responsible for the renewal of the epidermis. In the following spinous layer (stratum spinosum), the keratinocytes in normal conditions lose the ability to proliferate and start differentiating, strengthening the cellular mechanical properties and intercellular adhesions. In the granular layer (stratum granulosum), the keratinocytes become flatter and undergo terminal differentiation accompanied by further strengthening of the intercellular adhesion, structural cytoskeletal elements, and phospholipid synthesis—to then become anucleated corneocytes in the outermost corneal layer (stratum corneum) at the air–liquid interface [5]. Other cell types within the epidermis include melanocytes (which provide UV protection), Langerhans cells (specialized antigen-presenting dendritic cells), and Merkel cells (mechanoreceptors).

During inflammation, the cell populations in the skin undergo substantial changes, such as: pro-inflammatory cytokine production by the keratinocytes, fibroblasts, and immune cells; increased immune cell infiltration into the dermis and epidermis; keratinocyte hyperproliferation and improper differentiation; the loss of the intercellular permeability barrier; and increased ECM remodeling by the fibroblasts [5]. Numerous studies indicate that different cell types present in the skin are mechanosensitive. However, our understanding of the functional importance of mechanical cues and mechanotransduction in the pathogenesis of cutaneous inflammation is far from complete [6], and many mechanisms are only thought to be understood via extrapolation from studies on other experimental systems.

## 2. Inflammatory Skin Disorders

Inflammation of the skin can be induced by various external stimuli (UV radiation, allergen exposure, infection, contact with irritants, or mechanical stimuli), as well as due to intrinsic factors, e.g., autoimmune responses, inherited mutations, etc. Inflammatory skin disorders are characterized by the activation of the innate and/or adaptive immune system, the increased production of pro-inflammatory cytokines, and a shift of the gene expression and behavioral program in keratinocytes and fibroblasts to the wound-healing phenotype [7]. Depending on the cytokine signature, prevalence of specific immune cells and the extent of the reaction, different types of inflammation in the skin can develop.

Psoriasis is a common inflammatory disease with a prevalence of 2–3% in the population. It is a T-helper 17 (Th17)-driven disease that manifests with a severe chronic inflammation without a fibrotic component (Figure 1B). The major cytokines involved are interleukins IL-17, IL-22, and IL-23 [8]. Psoriasis is primarily characterized by perturbed homeostasis of the epidermis that results in epidermal hyperproliferation, aberrant differentiation, and disrupted barrier function, leading to the formation of red scaly plaques on the skin. Individuals with psoriasis are at an increased risk of developing other chronic and serious health problems such as psoriatic arthritis, metabolic syndrome, depression, and cardiovascular diseases, substantially contributing to the patients’ increased morbidity and mortality [9,10,11]. Many features of the psoriasis pathology resemble inflammatory and proliferative phases of wound healing that fail to resolve, which in fact facilitates the reparation of injury within the psoriatic plaques. At present, many aspects of the molecular pathology of keratinocytes and the underlying mechanisms that trigger psoriasis remain elusive. The frequent manifestation of the disease at the extensor sites of joints alongside the Koebner phenomenon [12] is considered a clue towards mechanical stress being a relevant pathogenic factor.

Atopic dermatitis (AD), or eczema, is a very common inflammatory condition characterized by skin redness, itching, extensive barrier dysfunction, and a prominent T-helper 2 (Th2) cytokine signature with upregulated IL-4 and IL-13, whereas in chronic eczema, interferon (IFN)-γ, IL-17, and IL-22 become more important [13,14].

While many skin inflammatory conditions do not result in tissue fibrosis or scarring, skin fibrosis due to the excessive deposition of ECM components in the dermis is a hallmark of systemic sclerosis (scleroderma). Scleroderma is a rare but very severe disease characterized by autoimmune activation, vascular damage, and the excessive deposition of ECM proteins such as collagen and fibronectin, leading to the loss of function of multiple organs, with increased skin stiffness being one of the main factors affecting the patients’ life quality [15,16,17] (Figure 1C). The altered physical properties of the ECM in fibrosis have been shown to upregulate mechanotransduction pathways [18].

## 3. ECM and the Mechanical Environment in the Skin

To maintain the structural integrity of the tissue, cells dynamically control ECM mechanics through the synthesis, organization, and degradation of its components. The composition, structure, and mechanics of skin ECM have been well described elsewhere [19,20,21]. The abundance, orientation, and crosslinking of different ECM fibers affects the ECM’s density, elasticity, and permeability for cells and soluble mediators [22]. ECM provides a structural basis for cell migration and appears to be the major factor defining tissue stiffness. A manifold increase in tissue rigidity has been observed in multiple human pathologies, such as fibrosis and cancers that have been associated with increased cell motility [23,24,25]. In psoriasis, the lesional skin exhibits a lower elasticity and a higher viscoelastic-to-elastic ratio compared with the adjacent uninvolved skin [26,27].

Fibroblasts are the principal cells that produce, organize, and reabsorb ECM. They (and potentially other cell types) can be activated by inflammatory cytokines, such as the transforming growth factor β (TGF-β), and differentiate into myofibroblasts characterized by extensive ECM production; an α-smooth muscle actin-positive cytoskeleton that is able to contract ECM fibers; enhanced cell migration; and the overexpression of adhesion molecules, cytokines, growth factors, and their receptors [28,29]. Myofibroblasts appear transiently during normal wound healing, but are consistently found in fibrotic diseases such as scleroderma. Increased ECM stiffness, previously regarded as the endpoint of fibrosis, is now seen as a critical factor driving fibrogenesis through the pathological upregulation of mechanotransduction that steers the cells away from the normal wound-healing program [3,30]. Stiffness-mediated mechanotransduction promotes further fibroblast activation, collagen deposition and crosslinking, cell proliferation and survival, and profound changes in the gene expression [31,32], therefore establishing a positive fibrotic loop [3,18].

Whereas psoriasis and many other cutaneous inflammatory diseases are not associated with skin fibrosis, accumulating data show that multiple ECM components, such as fibronectin, tenascin-C, and periostin, are upregulated or dislocated in the skin of psoriasis and AD patients and may contribute to keratinocyte hyperproliferation and the production of pro-inflammatory cytokines [33,34,35,36,37,38,39].

Tenascin-C is expressed during development, wound healing, and in pathological conditions, including chronic inflammation and cancer [40]. Tenascin-C is overproduced in an inflammatory dermis and can induce pro-inflammatory pathways via Toll-like receptor 4 (TLR4) signaling [41]. Its transcription has been shown to increase in response to cell tension, but at the same time tenascin-C can reduce cellular interactions with other ECM proteins, decrease Rho-GTPase activity, and decrease the contraction of collagen gels by the cells [40,42]; therefore, its pathogenic role is ambiguous.

Two types of fibronectin—soluble plasma fibronectin (which is produced by the hepatocytes into circulation and incorporates into fibrin clots upon injury) and insoluble cellular fibronectin (a component of the ECM)—are implicated in the pathogenesis of skin inflammation. An abnormal presence of plasma fibronectin has been detected in the basal layer of the epidermis in the nonlesional skin of psoriatic patients, making the keratinocytes susceptible to T cell cytokine-driven hyperproliferation [43]. The production of the extra domain-positive (EDA+) form of cellular fibronectin—a marker of angiogenesis and tissue remodeling—appears to be upregulated in psoriatic skin [37] and in scleroderma. It can promote fibroblast differentiation via integrin α4β7 and the MAPK/Erk1/2 pathway [44] and stimulate collagen production through TLR4 signaling [45]. Fibronectin has also been shown to redistribute to the corneal layer of the epidermis in AD, facilitating bacterial adhesion and promoting the infection of Th2 cell-mediated inflammatory skin lesions [46].

Periostin is expressed in the dermis, BM, and hair follicles during development and, to a lesser extent, in healthy adult skin, and is an important factor in stem cell fate regulation [47]. Periostin expression in the fibroblasts and basal keratinocytes can be induced by TGF-β and is upregulated in the wound-healing process. Periostin has been reported to be highly abundant in AD skin, contributing to itch, but is strongly reduced in psoriasis [48,49].

The synthesis of another dermal ECM component, collagen, has been also found to be upregulated, whereas its macromolecular properties deteriorate under chronic inflammation [50,51,52]. Increased collagen I production by fibroblasts in response to the eosinophil-produced leukotriene C4 has been connected to skin thickening in AD [53].

Skin inflammation substantially affects the organization of the epidermal BM [54]. In psoriasis-involved skin, the BM appears to be highly unstructured, exhibiting gaps and excessive folding, with a reduction in collagen IV but elevated levels of laminin and fibronectin [38,48]. In AD, the BM becomes thinner, with decreased levels of collagen IV and fibronectin, and increased hyaluronan and matrix metalloproteases (MMPs).

The alteration of the expression and activity of MMPs and ADAM metalloproteases that regulate ECM degradation is another key feature of pathological ECM remodeling [21,30,55]. Moreover, the ECM is a rich reservoir for cytokines, growth factors and other bioactive molecules that can be released by proteolysis via MMPs [55]. An overexpression of collagen-degrading MMPs (MMP2) by suprabasal keratinocytes, macrophages, endothelial cells, and fibroblasts has been reported in psoriasis patients [56,57].

Clearly, the role of different ECM components in inflammatory responses is not understood fully, and likely involves both positive and negative feedback mechanisms. The described changes in the ECM composition and physical properties (Table 1) can alter the mechanosignaling pathways in different cell types in the skin as well as provoke the infiltration of T cells and macrophages, in part because of the upregulation of cell motility due to increase of the substrate stiffness.

## 4. Integrins: Mechanosensitive ECM Receptors

Integrins represent a major class of ECM receptors in which unique combinations of α and β subunits create specificity for a variety of ECM molecules as well as a specific intracellular response [58]. Integrins function as mechanosensors: they activate and cluster, strengthening the adhesion when the base-level intracellular pulling force of the attached actin–myosin cytoskeleton encounters the resistance of the substrate. The increased mechanical load promotes the assembly of the integrin adhesions and the associated actin–myosin cytoskeleton, as well as the recruitment of signaling molecules, thus increasing intracellular mechanosignaling from the adhesion sites [59]. Integrin activation initiates multiple proliferatory and pro-survival pathways inside the cell: the MAP kinase cascade, c-Myc, c-Jun, and the Rac- and Ras-mediated pathways [60].

There is ample evidence that the disruption of skin homeostasis is associated with the dysregulation of certain integrins in the epidermis, such as the α2β1, α3β1, α5β1, α6β4, and αvβ6 integrin complexes [19,61]. This dysregulation might be a result of the inflammation-induced changes in the differentiation program and the upregulation of cell proliferation. At the same time, the alteration in the integrin expression itself has been shown to play a substantial role as a pathogenic factor in chronic skin inflammation.

The major integrins expressed in the epidermis are α2β1, α3β1, and α6β4, whereas other integrins are normally expressed at lower levels [61]. Integrin α6β4 primarily serves the purpose of BM attachment via hemidesmosomes in the interfollicular epidermis. Integrin β1 is enriched in epidermal stem cells that have a stronger adhesion to the BM than other progenitor cells [62], and its expression is normally restricted to the basal layer, being associated with the proliferative state of keratinocytes [43,62,63,64].

Integrin α5β1 becomes upregulated both in the basal and suprabasal keratinocytes during epidermal wound healing and in psoriatic inflammation [64,65,66,67,68]. The other integrins α2β1, α3β1, and α6β4 have also been shown to be abnormally expressed in psoriatic keratinocytes [65]. The suprabasal expression of integrin β1 alone or together with α2 and α5 in experiments with transgenic animals resulted in the development of psoriasis-like phenotypes [66,69]. As a result of alterations in the integrin expression profile and connection to the BM, dysregulated and/or misplaced integrin-dependent signaling can lead to keratinocyte hyperproliferation and disturbed epidermal homeostasis, which can then promote the production of cytokines and the onset of inflammation [66,70]. The integrin functional imbalance can also potentially create hypersensitivity to mechanical or inflammatory stimuli in the epidermis. For example, β1 integrin-positive keratinocyte stem cells in the nonlesional skin of psoriatic patients were found to be hyperresponsive to T cell cytokines [43].

The migration of immune cells to the sites of inflammation is also dependent on the altered dermal and BM ECM and on the integrin expression in the skin vasculature and immune cells [71,72]. For instance, integrin α1β1 is exclusively expressed by epidermal, but not dermal, T cells and is crucial for the T cell migration and accumulation in the epidermis in psoriasis pathogenesis [71]. Alterations in the endothelial integrin expression were also reported in psoriasis: integrin avβ3 was upregulated in the lesional vasculature and integrin β4 was decreased in the lesional superficial dermal vessels [73]. The suppression of certain integrins can inhibit leukocyte migration to the sites of inflammation, and may therefore represent a promising therapeutic strategy [74,75,76].

In addition to the function of intracellular mechanosignaling in adhesion and mediation, integrins can be involved in cytokine activation. Integrin αv complexes have a unique function in activating TGF-β, which represents a crucial soluble factor activated in wound healing and is involved in the promotion of fibrosis by inducing myofibroblast differentiation [4]. TGF-β, which is secreted in latent form and bound to ECM through a latency-associated peptide (LAP) [21], can be released and activated by mechanically stretched integrin αv complexes (such as αvβ5, αvβ6, and αvβ1) in response to an increased ECM stiffness. Increased integrin αv expression has been found to be associated with inflammation and the fibrotic process of different organs [77,78]. Integrin αvβ5 and αvβ6 are expressed in the proliferative basal layer of the normal epidermis, and become upregulated in migrating cells and in chronic wounds [67,79]. In the dermis, increased αvβ3 and αvβ5 integrin expression by fibroblasts can contribute to fibrosis by reinforcing the TGF-β-driven positive fibrotic loop [80,81]. Transgenic mice that overexpressed the β6 integrin, which pairs with αv in the basal keratinocytes, developed spontaneous chronic wounds and fibrosis, whereas an integrin-modulating therapy using anti-αvβ3 and αvβ5 antibodies prevented fibrosis and autoimmunity in the murine model of scleroderma [82].

Thus, the expression of most integrin subunits is affected in inflammatory conditions and fibrosis in different cell types in the skin (Table 2), and integrin-modulating therapies have been proposed as a potentially promising route for treating various inflammatory disorders [74,75,76,82].

## 5. Epidermal Cell–Cell Adhesions

Cell–cell adhesions in the epidermis include adherens junctions (AJs), tight junctions (TJs), desmosomes, and gap junctions formed by various transmembrane proteins and the associated cytoskeletal systems inside the cell. They provide adhesion and semipermeable barriers between cells, which are crucial for the mechanical properties of the tissue and the functional integrity. Cell–cell adhesions serve as sensors and transducers of mechanical forces, and the molecular basis for these functions has been well described elsewhere [83].

AJs are global master regulators of cell–cell adhesion, and their disruption results in the disassembly of other cell junctional complexes. AJs are formed by mechanosensitive transmembrane cadherin receptors (E-cadherin in epithelial cells) that establish a homotypic interaction between two neighboring cell membranes [84], and intracellular adaptor proteins such as α-catenin, β-catenin, and p120 that link the adhesion to the contractile actin–myosin bundles. AJs strengthen under a mechanical load and represent a “hot spot” for mechanosensing and mechanotransduction [85]. Vinculin can be recruited to AJs by α-catenin in a tension-dependent manner, providing a strong physical basis for the transmitting forces between AJs and the cytoskeleton [86]. β-catenin has been proven responsible for the tension-induced engagement of AJs in the mouse epidermis [87].

Alterations of AJs in an inflamed endothelium and simple epithelia such as intestinal have been well documented, and are linked to the inflammation-induced increase in the paracellular permeability of the cell layers [88,89,90,91,92,93]. Despite that, the role of epidermal AJs in cutaneous inflammation in particular has not been well studied. The compromised skin barrier in psoriasis and AD is considered to be primarily caused by improper keratinocyte differentiation, sphingolipid synthesis, and the dysfunction of the stratum corneum [94,95,96].

At the same time, inflamed human skin shows impaired cohesion between keratinocytes (spongiosis), which manifests as a widening of the intercellular spaces and narrow, elongated intercellular bridges. This suggests alterations in the cell–cell adhesions, similar to other epithelia under inflammatory conditions [56,97]. The decreased expression of E-cadherin and the upregulation of another type of cadherin, P-cadherin, have been reported in psoriatic skin and dermatitis [98,99,100,101,102]. Changes in the cadherin function affect AJ integrity, the recruitment and activity of adaptor proteins, and therefore, the AJ-mediated mechanosignaling.

The AJ protein β-catenin is one of the key regulators of mechanotransduction pathways in different cell types. In particular, it mediates the canonical Wnt pathway and can shuttle between an AJ and the nucleus (when Wnt signaling is on), or be degraded in the cytoplasm. β-Catenin protein expression has been found to decrease in the keratinocytes of psoriatic skin in some studies [99], whereas others showed nuclear translocation of the β-catenin, indicating the activation of canonical Wnt signaling in the context of psoriasis and other inflammatory conditions [103,104,105,106]. Wnt/β-catenin signaling has been found to promote profibrotic gene expression and skin fibrosis in mice and to be upregulated in systemic sclerosis. Moreover, the activation of the Wnt pathway can induce autocrine TGF-β signaling and itself be enhanced by TGF-β, indicating that these pathways are mutually dependent [29,107,108]. The Wnt/β-catenin pathway might be modulated by the mechanical stiffness of the ECM; however, the specific mechanisms are yet to be elucidated [109]. Another AJ protein, p120-catenin, has been found to suppress pro-inflammatory cascades in mouse skin [110].

TJs are intercellular adhesion complexes in epithelia and endothelia with their main function being controlling the paracellular permeability. TJs take part in mechanotransduction, and are integrated into the network of adhesion complexes together with AJs and the focal adhesions to regulate the intracellular tensile forces [83,111]. TJs are formed by transmembrane proteins (protein crumbs homologue 3, occludin, and claudins) and adaptor proteins (zonula occludens proteins ZO1, ZO2 and ZO3; cingulin; PAR3, PAR6, etc.), which link TJs to the actin cytoskeleton and define the epithelial polarity and the associated intracellular signaling components (e.g., Rho GTPases). In the skin, TJs contribute to barrier formation in the granular layer of the epidermis [112]. The apical positioning of the TJs in a multi-layered epidermis (and therefore the formation of a proper barrier) has been linked to E-cadherin-mediated mechanotransduction [113]. In addition, TJ proteins have been shown to regulate the proliferation, differentiation, cell–cell adhesion, and apoptosis in keratinocytes. TJ protein expression is affected by various inflammatory cytokines, among which are IL-1β, TNF-α, and IL-17A, and the TJ structure and function undergo substantial changes as part of the inflammatory response [92,114,115]. A downregulation of the TJ protein expression has been reported in AD [115,116] and psoriasis [117]. Increasing evidence shows that this downregulation might itself be a pathogenic factor for cutaneous inflammation [13], primarily due to barrier dysfunction, whereas other mechanisms can also take place. For example, claudin-1 deficiency in mice increased the levels of IL-1β, IL-12, IL-10, and INF-γ [118].

Desmosomes are very strong cell–cell adhesions in epithelia and some other cell types. They are formed by desmosomal cadherins (desmoglein and desmocollin), linked in the cytoplasm to plakoglobin, plakophilin, and desmoplakin, and connected to intermediate filaments. They are able to withstand considerable forces, providing epithelial tissue stability under mechanical stress as well as regulating barrier function. Desmosomal protein expression and localization in keratinocytes can be also mechanically regulated [119]. However, it remains unclear whether desmosomes take a direct part in mechanotransduction responses [83]. Recent data indicate that the strain-induced junctional remodeling of E-cadherin, the actin–myosin cytoskeleton, and mechanosensory yes-associated protein (YAP) signaling in keratinocytes can be dependent on desmoglein Dsg3 [119]. The role of desmosomes in establishing a proper barrier and mechanosignaling from the cell–cell adhesion sites can be implicated in the development of skin inflammation [120]. In fact, genetically impaired barrier functions (for instance, due to Dsg1 mutations) cause severe dermatitis due to an increased epidermal permeability [120,121].

Gap junctions are semipermeable connexin-based channels between cells that mediate both electrical and biochemical signals and are able to regulate cell proliferation, migration, apoptosis, and immune responses [122]. Mechanosensitivity has been reported for some connexins (Cx43, Cx30, Cx46, Cx50) [123,124,125,126]. Cx43 is expressed in basal proliferating keratinocytes that regulate epidermal differentiation and barrier function [6]. Alterations in gap junction protein expression and function have been described in lymphatic vessel and inflammatory lung diseases, infections, liver injury and hyperproliferative skin disorders [122,127,128,129]; however, the exact role of gap junction proteins in inflammatory responses is not well defined and may be irrespective of their function in gap junctions [129]. For instance, Cx26 has been reported to increase its expression in keratinocytes during wound healing and in the epidermis of psoriatic patients, and promote an inflammatory response in resident immune cells, whereas epidermal Cx43 is downregulated in acute wounds [122,129].

Overall, the composition, distribution, mechanics, and downstream signaling of all types of cell–cell adhesions are globally affected in the epidermis, as well as in the endothelium during skin inflammation. In turn, these alterations modulate the differentiation, proliferation, cytokine production, and immune cell interactions in these cell types.

## 6. Actin–Myosin Cytoskeleton and Cell Contractility

The actin cytoskeleton is a very dynamic mechanosensitive machinery that is able to generate pushing and pulling forces within the cell and is essential for virtually all aspects of cell physiology: adhesion, migration, cytokinesis, membrane trafficking, secretion, stem cell pluripotency, and differentiation [130,131]. Nonmuscle myosin II (NMII) is an actin-dependent motor protein that is responsible for the majority of the force generated by the cell. In large part, the NMII contractility is applied to the adhesion-anchored actin filaments. Thus, the actin-based integrin and cadherin adhesions dynamically respond to the NMII-generated forces, and therefore act as the main hubs for mechanosignaling.

Overall, the roles of the actin–myosin cytoskeleton in inflammatory processes have been studied predominantly in the epithelia and stroma of the gastrointestinal and respiratory systems. To date, the actin–myosin cytoskeleton regulation during skin inflammation is not well described; however, some parallels can be drawn between simple columnar gut and airway epithelia and the keratinocytes in the stratified epidermis. Generally, the actin cytoskeleton and the level of NMII contractility directly impact the cell–substrate adhesion and the cell–cell junction integrity, and its perturbations can lead to a disrupted epidermal barrier [91] and decreased contact inhibition of proliferation [132], thus promoting pro-inflammatory pathways. In the tissue stroma, actin–myosin contractility forces are crucial for the rigidity sensing and ECM remodeling by fibroblasts, the regulation of blood vessel permeability by endotheliocytes, and immune cell motility and function [133,134,135,136]. In primary human keratinocytes, the actin cytoskeleton and focal adhesions are able to reorganize in response to the psoriatic cytokines IL-17A, IL-17E, and IL-9 [137,138], and may regulate the sensitivity of inflammasome activation [139,140].

Rho GTPase signaling pathways link the membrane receptors to the regulation of the actin cytoskeleton, associated adhesion complexes, and gene transcription to promote coordinated changes in cell behavior [141]. Rho activates Rho-associated kinases (ROCK1 and ROCK2), which phosphorylate and activate the NMII motor and formin mDia, a mechanosensitive actin-polymerizing protein [142,143]. The Rho-ROCK pathway has a direct involvement in cell responses to the substrate stiffness, migration during wound healing, neutrophil transmigration, endothelial and epithelial barrier function, and the balancing proliferation and differentiation in human keratinocytes [144,145,146,147]. The available data support the evidence for Rho-ROCK signaling involvement in the stimulus-induced disruption of AJs and TJs and the associated alterations in adhesion-mediated mechanosignaling [91]. Moreover, Rho-dependent actin polymerization can shift the abundance of the monomeric G-actin pool in the cytoplasm and nucleus, which is another factor for transcription regulation [148].

Myosin light chain kinase (MLCK) is another major kinase that activates the NMII motor. Growing evidence shows diversity in the ROCK and MLCK functions in the regulation of NMII activation [149,150]. MLCK induces a contraction of the actin–myosin belt associated with cell–cell junctional complexes, and an upregulation of MLCK expression has been observed in inflammatory disorders of multiple organs, accompanied by the loss of the transepithelial barrier [93]. In intestinal bowel disease, MLCK has been shown to mediate the intestinal barrier dysfunction through NMII-contractility-dependent TJ loss [151]. Moreover, MLCK hyperactivation triggers endothelial barrier disruption, increasing the extravasation of immune cells and the tissue permeability for the mediators coming from the blood. MLCK, therefore, may represent a potential target for the treatment of inflammatory disorders of various organs [151].

Based on a variety of its heavy chains, NMII is expressed as three isoforms: NMIIA, NMIIB, and NMIIC, which have both common and unique properties due to differences in their kinetics, load-dependence, and interacting partners [152,153]. Epidermal keratinocytes express all three isoforms and dermal fibroblasts express NMIIA and NMIIB. Even though the role of NMII isoforms has started to be recognized in the disease, the information on their involvement in skin inflammation is very scarce. Some evidence indicates that the aberrant expression or activity of NMII isoforms can be found in many infectious diseases [154]. In particular, NMIIA is required for T cell transendothelial migration [155]. Both NMIIA and NMIIB are necessary for efficient ECM remodeling, but have distinct roles in this process [156]. The NMIIA and NMIIB expression in fibroblasts is upregulated with increasing ECM stiffness, in correlation with an increased tissue remodeling by the cells [157]. NMIIB may be also implicated in the pathogenesis of fibrosis, as described in the lungs [158]. At the same time, therapeutic strategies based on NMII inhibition may face the problem of multiple side effects due to NMII’s universal functions, whereas targeting specific NMII kinases might be more appropriate [91].

The dysregulation of multiple actin-binding proteins can also play a role in the inflammatory processes in tissues [159]. For example, the actin crosslinking protein fascin is overexpressed in inflammatory bowel disease and is involved in immune cell locomotion and mechanotransduction through the nuclear envelope [160,161]. The recently discovered and poorly characterized myosin-18A may also play a role in inflammatory responses, presumably via the regulation of epithelial cell migration and the activation of natural killer cells in response to the elevation of surfactant protein A in psoriatic skin [162,163,164]. The roles of other actin-cytoskeleton-associated proteins, such as cofilin [165] and septins [166,167], in inflammatory responses are also beginning to emerge.

## 7. Intermediate Filaments

Intermediate filaments (IFs), previously considered as structures that primarily provide mechanical properties to the cells and tissues, now continually emerge as participants in signal transduction. The mechanical stimuli are transmitted to the IFs through the cell–cell junctions and cell–ECM adhesions, such as desmosomes and hemidesmosomes in epithelial cells. Moreover, IFs have been shown to connect to the actin-associated cadherin and integrin adhesion receptors in a force-dependent manner [168]. Their abundance in the cytoplasm, physical robustness, and ability to strongly deform and stretch under force provide the cell with a powerful system to “buffer” and withstand mechanical stresses [169,170,171] and possibility counterbalance the focal-adhesion mediated forces [172].

Keratinocytes express a variety of skin-specific keratin Ifs, and their expression patterns depend on the differentiation status of the cells [173]. For instance, the keratins K5 and K14 are expressed in the basal layer and the keratins K1 and K10 are expressed in the suprabasal layer of a normal interfollicular epidermis. The keratin IF network, particularly K14, can regulate keratinocyte rigidity sensing and nuclear mechanotransduction [174]. In inflammatory skin conditions, such as psoriasis, AD or ovalbumin-sensitized skin models in mice, the expression of the differentiation-related keratins K1 and K10 is downregulated, whereas keratins K6, K16, and K17 are induced [175,176,177]. The transcription of these “inflammatory” keratins can be promoted by the cytokine- and growth factor-activated pathways. For example, the transcription of keratins K6 and K16 is induced by the epidermal growth factor (EGF), the tumor necrosis factor α (TNF-α), and IL-1, and that of keratins K17 and K7 is induced by IFN-γ [176].

In turn, IFs can also regulate inflammatory responses. Keratin K17 promotes Th1 and Th2 inflammatory response and keratinocyte proliferation [178]. Roth and colleagues have demonstrated that the absence of keratin K1 causes a prenatal increase in IL-18 and the S100A8 and S100A9 proteins, and a skin barrier defect in mice [179]. In addition, the importance of proper IF function is also highlighted by the fact that mutations in the keratin genes cause various cutaneous diseases and a predisposition to inflammatory disorders [180].

The role of vimentin IFs, which are expressed in mesenchymal cell types, in inflammatory responses and the regulation of cell fate has begun to be recognized. The reorganization of the vimentin cytoskeleton is necessary for cell migration during the realization of the wound healing program, as well as chemokine-induced leukocyte migration to the inflammation sites and the regulation of the transendothelial barrier [181]. Vimentin deficiency in mice aggravates intestinal inflammation [182] and attenuates lung injury and fibrosis by acting as a platform for inflammasome assembly [183], whereas no data specific to skin inflammation are available to date.

In recent years, mechanoperception by the nuclear surface itself has been shown to modulate gene expression across the genome [184,185,186,187]. The mechanical signals are transduced to the nucleus from the cytoplasm via the linker of cytoskeleton and nucleoskeleton (LINC) complex, which is embedded into the nuclear envelope and linked to all cytoskeletal systems [188]. The transmitted forces induce modifications of nuclear lamins (nuclear IFs), which change the chromatin structure [148]. The reorganization of the nuclear lamina is associated with inflammation, including the activation of the inflammatory response genes [189].

Overall, the inflammation-induced changes in the IF proteins may result in the alteration of the skin’s mechanical properties as well as mechanosignaling [174], which can be induced by the IFs themselves or through the other cell structures that sense altered cell mechanics upon IF dysregulation.

## 8. Mechanosensitive Ion Channels

Mechanosensitive channels represent another group of signal transducers that is crucial for the regulation of skin homeostasis [190,191,192]. They activate in response to physical stretching of the cell membrane, and can mediate mechanical stress-induced calcium waves and intracellular calcium signaling in keratinocytes [193], which is essential for many aspects of skin physiology.

Transient receptor potential (TRP) channels are implicated in the regulation of keratinocyte proliferation and differentiation, skin barrier and immune functions [191,194], and the sensation of inflammatory pain [195]. Their specific expression and non-neuronal cell functions in normal and diseased skin have been recently reviewed in [196]. The canonical TRP cation channels TRPC1, TRPC4, and TRPC6 are key regulators of calcium-induced keratinocyte differentiation. The reduced expression of TRPC channels in psoriatic keratinocytes can disturb the calcium gradient in the epidermis, leading to the impaired differentiation and hyperproliferation of cells [197].

The activity of the TRP vanilloid subtype channels TRPV1, TRPV3, and TRPV6 in epidermal keratinocytes can suppress cell growth, have a negative effect on the skin barrier, and promote neurogenic skin inflammation [191]. TRPV4 has been shown to mediate stretch-induced actin polymerization, the activation of ROCK, and the upregulation of focal adhesion-mediated signaling [198], whereas its inhibition prevents the stretch-induced upregulation of the p38 MAPK cascade and the production of the pro-inflammatory cytokines IL-6 and IL-8 [199]. The activity of the TRP subfamily A member 1 (TRPA1) channel can regulate the expression of certain adhesion and ECM proteins and differentiation in keratinocytes [6]. TRPA1 was also found to enhance dendritic cell migration to the lymph nodes in rodent dermatitis models [6].

The Piezo family channels (Piezo1 and Piezo2) open in response to membrane stretching, thus transducing mechanical signals into nonselective cation flow. Piezo1 has been linked to inflammatory responses in various tissues [200]. For instance, Piezo1-mediated calcium signaling regulates macrophage polarization and the production of inflammatory mediators (IL-6, TNF-α, CXCL2, prostaglandin E2) in response to IFNγ/LPS stimulation [201,202]. Piezo1 activity is integrated with the actin cytoskeleton and adhesion structures; it can respond to NMII-generated traction forces [203]; promote α4, α5, and β1 integrin-dependent adhesion [204]; tether to the cadherin-β-catenin mechanotransduction complex [205]; and regulate keratinocyte migration [206]. The inhibition of Piezo1 impairs key functions of the fibroblasts and endothelial cells during wound healing [207]. Inflammatory signals can also enhance the mechanosensitive calcium influx mediated by Piezo2 [208], and in Merkel cells, Piezo2 has been shown to regulate the development of itch [209].

Therapeutic approaches targeting mechanosensitive channels can be exploited in the management of highly prevalent skin conditions such as AD, psoriasis, and acne [196,197].

## 9. Mechanoactivated Intracellular Signaling

The crosstalk between different types of adhesions, cytoskeletal structures, receptors, channels, and regulatory molecules results in the integrated signaling that defines the mechanical perception by the cell [148,210]. Examples of this crosstalk include the mutual regulation of epithelial junctions, Rho family GTPases and calcium receptors [211,212], and the ability of IF-linked hemidesmosomes and desmosomes to oppose force transduction by focal adhesions and AJs, respectively [172].

Numerous studies indicate that inflammatory stimuli activate multiple mechanosensitive signaling pathways (Figure 2). In skin inflammation, these pathways generally induce cell proliferation and de-differentiation in the epidermal keratinocytes. The inflammation-induced changes can be described as an activation of the “wound healing” phenotype and cytokine production in both the epidermis and dermis. In the immune cells, the activation of mechanotransduction can also increase cell motility and cytokine production. However, the specific mechanisms can vary depending on the type, severity, and duration of the inflammation, as well as the patient’s genetic background, etc. Malakou and colleagues have recently reviewed the mechanotransduction pathways that may be relevant to psoriasis pathogenesis [213]. We are often still missing data linking certain pathways to specific inflammatory conditions, and their relevance in one system cannot always be projected onto another. Here, we discuss several key mechano-dependent components that are relevant for skin inflammatory processes.

Focal Adhesion Kinase (FAK) is a signaling molecule recruited to the ECM adhesion sites from the cytoplasm and phosphorylated in response to the mechanical load. Its activity appears to be required during mouse skin development, but dispensable for skin homeostasis [214]. The FAK-p38-Myc pathway can be activated by integrin αv and is necessary for 3D skin formation [215]. FAK itself can control the transcription of multiple genes in the nucleus, including chemokines [216], as well as enhance inflammasome activation [217]. Nowell and colleagues showed that the FAK inhibitor PF562271 can suppress corneal metaplasia in response to inflammation in the mouse model, pointing to its pro-inflammatory effects [105]. FAK hyperactivation has been explicitly linked to cancer progression and fibrosis pathology, and its inhibition can abrogate the TGF-β-induced fibrotic phenotypes in human and murine fibroblasts [81,218,219,220]. However, the specific role of FAK in the keratinocyte function during skin inflammation has not been demonstrated.

NFκB (nuclear factor kappa-light-chain-enhancer of activated B cells) is a highly conserved transcription factor that regulates cell growth, apoptosis, and inflammatory responses. The NFκB-activating pathways can be induced by various cytokines, inflammatory molecules, and stress signals [221]. Multiple studies have confirmed the importance of NFκB signaling in epithelial tissue homeostasis, and particularly in skin immunity [222]. Apart from its immune function, NFκB has been identified as a mediator of mechanotransduction. NFκB nuclear translocation has been observed in different cell types in response to mechanical stress [223,224]. The NFκB pathway can be activated downstream of focal adhesion and FAK-mediated mechanical (in contrast to TNF-mediated inflammatory) signaling [225], and is coordinated by Rho-GTPases [226]. Moreover, a disruption of the actin cytoskeleton, loss of cell–cell contacts, or p120 deficiency in epithelial cells induces NFκB activation and the downstream production of inflammatory mediators [110,227,228].

The serum response factor (SRF) and its cofactor MRTF are ubiquitous transcription regulators, particularly for differentiation- and adhesion-related genes [229]. The SRF mediates various cellular mechanoresponses, and its activity is essential for the skin development and homoeostasis in mice and humans. The SRF has been shown to promote substrate adhesion-regulated terminal differentiation of keratinocytes acting downstream of RhoA and actin polymerization [230]. SRF/MRTF translocation to the nucleus can also be regulated via contractility-dependent E-cadherin-mediated signaling at the AJs [229,231]. Besides the transduction of physical cues from the cell–cell and cell–ECM-linked cytoskeleton to the nucleus, the SRF is responsible for feedback cytoskeletal modulations such as spindle orientation and cell adhesion. The SRF is required for actin polymerization and integrin adhesion in neutrophils during their migration in response to inflammation [232]. SRF deficiency in mouse skin results in epidermal hyperplasia, the aberrant expression of differentiation markers and transcriptional regulators, and enhanced inflammation [233].

The transcription co-activator YAP and its homolog TAZ take a critical role in mechanotransduction as sensors and mediators of the mechanical cues of the cellular microenvironment via a Rho-dependent mechanism [234]. Hippo pathway-regulated YAP signaling has been proven critical for the size control of various organs. For instance, the integrity of epithelial cell–cell junctions and an increased cell density inhibited YAP signaling. which limited cell proliferation and tissue expansion [235,236]. The Roles of YAP/TAZ and the Hippo pathway in epidermal stem cell regulation and skin homeostasis were recently reviewed in [237]. An increased nuclear localization of YAP/TAZ and a perturbed cell differentiation was found to be associated with chronic inflammation in the corneal epithelium [105] and psoriatic inflammation [237,238]. The inhibition of NMII contractility and the subsequent loosening of AJs in cultured keratinocytes, on the contrary, promoted YAP nuclear entry [132]. IL-17A, one of the main drivers of psoriatic inflammation, is able to activate YAP pathway in keratinocytes in vitro [239]. YAP/TAZ upregulation has been associated with the pathogenesis of systemic sclerosis [240,241]. Rho-dependent YAP activation by diverse ligands of the G-protein-coupled receptors (GPCR) might act in synergy with TGF-β1 to reinforce the pro-fibrotic program in human dermal fibroblasts, whereas the inhibition of YAP/TAZ limits the TGF-β signaling, myofibroblast differentiation, and skin fibrosis in mouse models [31,241]. Moreover, the crosstalk between YAP/TAZ, NFκB, and β-catenin signaling may have complex positive and negative effects on inflammatory responses [242,243].

Activating protein-1 (AP-1), a dimeric transcription factor composed of Jun and Fos proteins, is an essential global regulator of YAP/TAZ-dependent gene expression for the regulation of the proliferation, differentiation, survival, and migration of cells [244]. AP-1 has been shown to activate by mechanotransduction signaling pathways mediated via integrins, FAK, Rho, SRF, calcium influx, and MAPK/ERK1/2 to mediate pro-survival/proliferative responses. Piezo1 has been shown to lead to AP-1 activation, promoting the production of pro-inflammatory mediators in the innate immune cells in lungs [201]. In psoriasis, AP-1 mediates CCL2 and IL-23 expression by dendritic cells [245]. Other studies indicate an anti-inflammatory role of the AP-1 subunits in cutaneous pathology; an abrogation of JunB/AP-1 in keratinocytes triggered chemokine and cytokine expression [246], and the Jun subunit expression was found downregulated in psoriasis vulgaris patients [247].

In addition to numerous mechanoregulated signaling proteins, a large number of mechanosensitive microRNAs are able to regulate cell behavior. The microRNA-dependent regulation of translation can buffer the expression of the proteins involved in mechanotransduction to maintain mechanical homeostasis of the tissue. Upregulation of certain microRNAs has been associated with chronic inflammatory conditions, including psoriasis and other skin disorders [6,248,249].

## 10. Skin Mechanical Stretch and the Koebner Phenomenon

Mechanical stimuli of a large scale, such as scratching and stretching, impact skin function all the time. In cutaneous pathology, a well-known example of the relevance of mechanical factors to inflammation pathogenesis is the Koebner phenomenon. It manifests as the appearance of new skin lesions following trauma or mechanical stress, which are uncharacteristic to the type of the trauma but clinically and histologically identical to the patient’s underlying cutaneous disease [12,250]. For instance, a “Koebner-positive” patient with psoriasis will develop psoriasiform lesions along the site of a trivial skin injury [250]. Whereas several theories have been proposed in attempt to explain the Koebner phenomenon, including immunological, vascular, dermal, infectious, and other factors, its molecular mechanisms remain unclear [250,251]. Indeed, its understanding poses a complex problem, as mechanical stimulation affects multiple cell types and structures in the skin simultaneously and can trigger multiple signaling pathways [252]. Experimental studies that utilize cell-stretching devices might shed light on some of the underlying mechanisms of this phenomenon. The mechanical stretching of human keratinocytes cultured on elastic silicone surfaces can induce proliferative phenotypes, the phosphorylation of EGFR and ERK1/2, the activation of Akt and anti-apoptotic pathways, the upregulation of the inflammation-related keratin K6, and the suppression of the keratinocyte differentiation marker K10 [253,254]. In another study, lateral stretch and squeeze of the epidermis induced the Rac-PAK1 pathway, leading to a strengthening of the epithelial cell attachment to the BM via hemidesmosomes [255]. A cyclic stretch of high magnitude has also been shown to boost the production of IL-6 [256], PGE2, COX-2, and MMP-1 gene expression [257] in human tendon fibroblasts and an enhanced expression and secretion of IL-8 and MCAF/MCP-1 via an actin-dependent mechanism in endotheliocytes [258].

The fact that mechanical stress by itself can trigger inflammation in some individuals implies that the nonlesional areas of the skin are already preconditioned to develop inflammation upon mechanical perturbation. This hypersensitivity can be potentially explained by the upregulated base-level mechanosensation of keratinocytes and possibly other cell types. In the previous chapters, we attempted to describe the alterations of numerous cellular components that could potentially lead to this hypersensitivity, as illustrated in (Figure 3). Future studies should provide a stronger link between the underlying biological traits and this intriguing clinical phenomenon.

## 11. Conclusions

Here, we reviewed the diverse mechanisms by which cells in the skin sense their biophysical environment to initiate a wide range of responses, including cell survival, proliferation, migration, and differentiation. The dynamic integration of common and cell type-specific mechanosensitive pathways contributes to the maintenance of the skin tissue homeostasis in normal conditions [210]. Many of these pathways become upregulated (and some downregulated) to adjust skin functions as part of the wound-healing program, infection interactions, or large-scale mechanical adaptations. At the same time, the dysregulation of mechanotransduction mechanisms significantly affects the onset and progression of inflammatory disorders and sometimes becomes a triggering factor in the development of skin pathologies.

Future studies should provide a systematic picture of the molecular mechanisms connecting the skin mechanics and inflammation, and focus on the mechanical crosstalk between different cell types in the epidermis, dermis, and subcutaneous skin layers during tissue repair, inflammation, and fibrosis [259].

## Figures and Tables

**Figure 1 cells-11-02026-f001:**
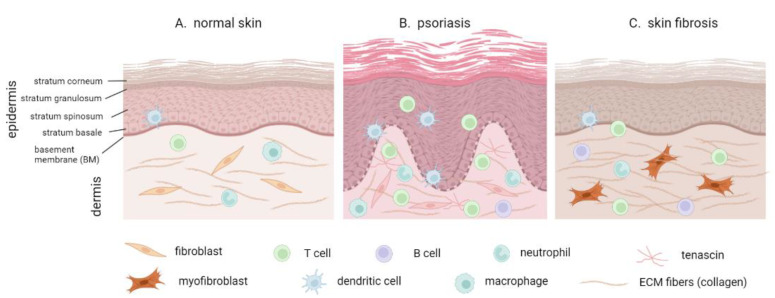
A scheme of the epidermis and dermis in normal skin (**A**), psoriasis (**B**) and fibrosis (**C**). Morphologically, psoriatic inflammation is characterized by epidermal hyperproliferation and thickening (acanthosis), an increased infiltration of immune cells into the dermis and epidermis, alterations in the composition of the dermal ECM, and damage to the BM. Fibrotic skin is characterized by excessive ECM deposition by the dermal myofibroblasts and an increased presence of immune cells in the dermis. Figure created in Biorender.com 14.06.2022.

**Figure 2 cells-11-02026-f002:**
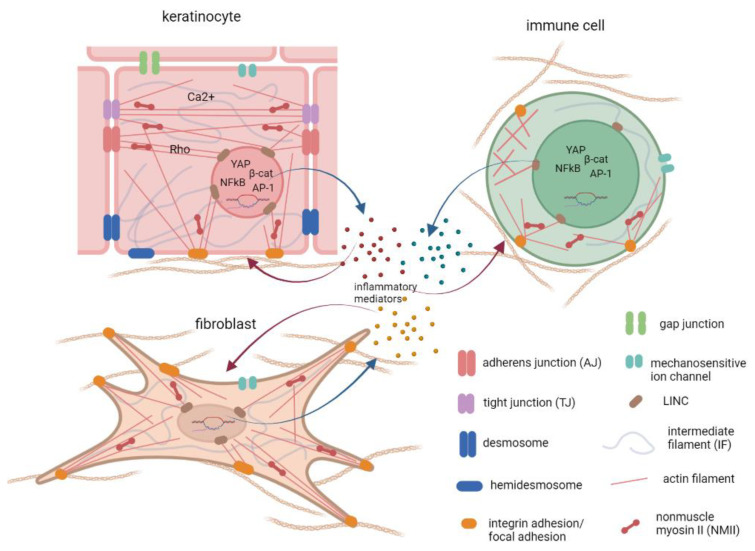
Integration of different mechanotransduction mechanisms during inflammation in skin cells. The cell senses forces and ECM structures via engagement of channels and adhesion receptors, which physically and biochemically connect to the cytoskeletal systems inside the cell. The alterations of the intercellular signaling pathways, together with the mechanical load on the nucleus, adjusts the gene expression. Figure created in Biorender.com 14.06.2022.

**Figure 3 cells-11-02026-f003:**
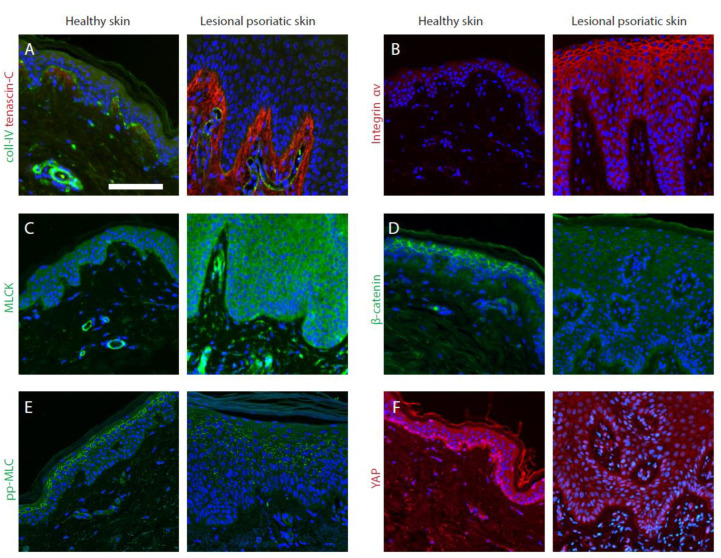
Dysregulation of mechanotransduction-related proteins in psoriatic skin revealed by immunofluorescence. (**A**) Collagen IV is generally lost from the BM and tenascin-C is overexpressed in the dermis of psoriatic skin. (**B**) Integrin αv is upregulated in the dermis. (**C**) In normal skin, MLCK has higher expression in the basal layer of the epidermis, whereas in psoriatic skin, it is increased in all epidermal layers. (**D**) β-Catenin is downregulated at the epidermal cell junctions. (**E**) The phosphorylated myosin light chain (Thr18/Ser19) underlies the cell borders in the upper epidermal layers in normal skin and the continuous pattern is lost in psoriatic epidermis. (**F**) YAP localizes to the nucleus in the basal layer, but not in the upper layers in the normal epidermis, whereas its nuclear localization is retained in multiple layers of a psoriatic epidermis. DAPI is in blue. Scale bar, 100 μm.

**Table 1 cells-11-02026-t001:** Skin ECM protein alterations during inflammation. For additional information on the immunomodulatory functions of the ECM, see ref. [38].

ECM Protein	Location	Inflammation-Associated Changes	Significance for Pathology	References
Collagen I and III	Dermis	Increased in chronic inflammation and fibrosis	Increased skin stiffness and integrin signaling in fibroblasts	[21,30,50,51,52,53]
Collagen IV and VII	BM	Disorganization of the fibrils; decreased expression in psoriasis	Increased dermal–epidermal permeability; affects integrin signaling in basal keratinocytes	[38,51,52]
Fibronectin (cellular)	Dermis, BM, epidermis	Increased in dermis during fibrosis; found in epidermis in AD	Increased skin stiffness and integrin signaling in fibroblasts; can promote bacterial colonisation in AD epidermis	[37,46]
Fibronectin (EDA+ cellular)	Dermis	Increased in inflammation (psoriasis) and fibrosis	Angiogenesis; tissue remodelling; fibroblast differentiation	[35,37,44,45]
Fibronectin (plasma)	Epidermis	Accumulates during inflammation (psoriasis)	Signaling and proliferation in basal keratinocytes	[43]
Laminin	BM	Overexpression; disorganized structure (psoriasis)	Increased immune cell adhesion; keratinocyte proliferation	[21,38]
Periostin	Dermis, BM	Increased expression during inflammation (AD) but not in psoriasis	Modulates dermal cell adhesion and signaling; stimulates cell proliferation; contributes to itch	[21,36,47,48,49]
Tenascin	Dermis	Increased expression during inflammation (AD, psoriasis)	Modulates dermal cell adhesion and signalling; stimulates cell proliferation; induces pro-inflammatory pathways	[21,33,34,40,41,42]

**Table 2 cells-11-02026-t002:** Integrin alterations during skin inflammation.

Integrin Complex	Ligand	Expressing Cells	Inflammation-Associated Changes	Significance for Pathology	References
α1β1	Collagen	Epidermal T-cells, fibroblasts, endothelial cells	Increased T cell and macrophage recruitment to inflammation sites	Pro-inflammatory effects	[71,72]
α2β1	Collagen	Keratinocytes, fibroblasts, endothelial cells	Upregulated in all epidermal layers in wound healing and psoriasis	Increased keratinocyte proliferation and altered differentiation	[65,70]
α3β1	Laminin	Keratinocytes	Upregulated in all epidermal layers in wound healing and psoriasis	Increased keratinocyte proliferation and altered differentiation; possible regulation of fibrosis	[65,70,80]
α5β1	Fibronectin	Keratinocytes (low in homeostasis), dermal fibroblasts, endothelial cells	Upregulated in basal and suprabasal keratinocytes during wound healing and in psoriasis; upregulated in fibroblasts under TGF-β	Fibroblast activation; increased keratinocyte proliferation and altered differentiation	[64,65,66,67,68,70,81]
α9β1	Tenascin	Keratinocytes (low in homeostasis), endothelial cells	Upregulated during wound healing	Increased keratinocyte proliferation and altered differentiation	[61]
α6β4 (hemidesmosomal)	Laminin	Basal keratinocytes, endothelial cells	Upregulated in psoriatic epidermis; decreased in psoriatic vasculature	Increased keratinocyte proliferation and altered differentiation	[65,70,73]
αvβ3	Vitronectin, fibronectin	Endothelial cells, fibroblasts	Increased in psoriatic skin vasculature	Pro-inflammatory effects	[73]
αvβ5	Vitronectin	Fibroblasts, keratinocytes (low expression)	Upregulated in fibrosis	TGF-β activation; pro-fibrotic effects	[61,80]
αvβ6	Fibronectin, tenascin	Keratinocytes (low in homeostasis)	Upregulated during wound healing	TGF-β activation; pro-fibrotic effects	[61,79]
αvβ8	Vitronectin	Suprabasal keratinocytes, dermal fibroblasts	Upregulated in fibrosis	TGF-β activation; pro-fibrotic effects	[61,80]

## Data Availability

The original data presented in this study are available on request from the corresponding author.

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
