# Peer review of "Mechanotransduction in Skin Inflammation"

_cells, 2022, doi:10.3390/cells11132026_

Round 1

Reviewer 1 Report

In the 15-page review “Mechanotransduction in skin inflammation”, based on 259 publications listed as reference materials, the authors systematically, albeit briefly, reviewed the current level of knowledge in the indicated area. In the introduction, the readers are refreshed on the structure and organization of the skin layers and informed that different types of skin cells are mechanosensitive. In the following chapters, the authors describe the main inflammatory skin disorders and explain a potential role in mechanotransduction of various components of the extracellular matrix, integrins, cell-cell adhesions, actomyosin and other components of the cytoskeleton, ion channels, and signaling cascades. The review is written clearly and concisely. The material is presented in a logical way and is easy to follow. Hypotheses are well separated from factual information and reasonably well justified. The gaps in knowledge are appropriately highlighted. Overall, I believe the review will benefit a broad community of researchers interested in mechanosensing mechanisms behind normal and diseased skin and its components.

I noticed only minor deficiencies and typos in the manuscript that are summarized below:

1.       Please provide a list of abbreviations.

2.       82: The second chapter is double marked (2. 2.)

3.       85-89: I am not sure that the use of numerous conjunctions “and” is justified in this sentence.  

4.       133: Can you briefly elaborate on the reorganization of ECM fibers by the contractile cytoskeleton?

5.       The opposition of cellular fibronectin to plasma fibronectin may help to clarify the term “cellular”.

6.       230: Expression of α1β2 integrin on what cells is meant in this sentence? Please clarify.

7.       241: Please check the spelling TGFβ vsTGF- β (and other similar terms) throughout.

8.       A somewhat more detailed explanation of the Koebner phenomenon (e.g., lesions as uncharacteristic for the type of the trauma) would be desirable for clarity.

9.       Please provide more detailed legends to each of the three figures. Alterations on Figure 3 is the least self-explanatory.

110.  648: The last sentence of the manuscript is not finished.

Author Response

In the 15-page review “Mechanotransduction in skin inflammation”, based on 259 publications listed as reference materials, the authors systematically, albeit briefly, reviewed the current level of knowledge in the indicated area. In the introduction, the readers are refreshed on the structure and organization of the skin layers and informed that different types of skin cells are mechanosensitive. In the following chapters, the authors describe the main inflammatory skin disorders and explain a potential role in mechanotransduction of various components of the extracellular matrix, integrins, cell-cell adhesions, actomyosin and other components of the cytoskeleton, ion channels, and signaling cascades. The review is written clearly and concisely. The material is presented in a logical way and is easy to follow. Hypotheses are well separated from factual information and reasonably well justified. The gaps in knowledge are appropriately highlighted. Overall, I believe the review will benefit a broad community of researchers interested in mechanosensing mechanisms behind normal and diseased skin and its components.

We thank the reviewer for the positive evaluation of our manuscript.

I noticed only minor deficiencies and typos in the manuscript that are summarized below:

  1.           Please provide a list of abbreviations.

The list of abbreviations is now provided at the end of the manuscript.

  1.           82: The second chapter is double marked (2. 2.)

Corrected.

  1.           85-89: I am not sure that the use of numerous conjunctions “and” is justified in this sentence. 

Corrected.

  1.           133: Can you briefly elaborate on the reorganization of ECM fibers by the contractile cytoskeleton?

Sentence corrected to avoid misunderstanding. We now saw that the ECM fibres can be contracted by the cytoskeleton, whereas reorganization would be more suitable in the context of producing ECM-crosslinking and degrading enzymes.

  1.           The opposition of cellular fibronectin to plasma fibronectin may help to clarify the term “cellular”.

Clarified.

  1.           230: Expression of α1β1 integrin on what cells is meant in this sentence? Please clarify.

Corrected.

  1.   241: Please check the spelling TGFβ vsTGF- β (and other similar terms) throughout.

Corrected.

  1.   A somewhat more detailed explanation of the Koebner phenomenon (e.g., lesions as uncharacteristic for the type of the trauma) would be desirable for clarity.

Added to the corresponding chapter.

  1. 9.       Please provide more detailed legends to each of the three figures. Alterations on Figure 3 is the least self-explanatory.

The figure legends were extended.

  1.   648: The last sentence of the manuscript is not finished.

Corrected.

Reviewer 2 Report

The paper is a review on the key players in mechanotransduction  within skin inflammatory diseases (psoriasis) and fibrosis as a result of chronic inflammations. The paper is well presented although tables to summarise the context of the paragraph would be of advantage. Moreover advantages and drawbacks should be more highlighted.

The paper presented is a review of the mechanotransductive key players in the context of skin inflammatory diseases (i.e. psoriasis) and skin fibrosis as a result of chronic inflammation. Uncovering the functional role of mechanical cues and mechanotransduction in the pathogenesis of cutaneous inflammation is of fundamental importance. The review appears not too in depth of the works they cover.    1) Comparison of all ECM components in terms of their inflammatory responses, advantages and disadvantages should be given. Studies on these components (related to stiffness of materials, cells, pathways, ECM components and everything else that the authors consider useful should be reviewed in more detail, compared and criticised. It should be taken out in (possibly) table/s.   2) For example, Line 204: “There is ample evidence that the disruption of skin homeostasis is associated with the dysregulation of certain integrins in the epidermis (19, 61).”Which integrins? Explain   3) On page 5 line 229 claim: “Migration of immune cells to the sites of inflammation is also dependent on the altered ECM and integrin expression in the skin (71, 72)”. Could the authors explain what exactly is altered?   4) It is worth adding a short introduction/conclusion at the end or beginning of each chapter describing why this kind of components is promising to the study carried out. As well as disadvantages and limitations of works carried out.   5) More criticism in the conclusions about all the work carried out so far and what message the authors want to send would be helpful.  

So, the work should be systematized, which will significantly improve the level of readers perception of this review. 

Author Response

The paper is a review on the key players in mechanotransduction  within skin inflammatory diseases (psoriasis) and fibrosis as a result of chronic inflammations. The paper is well presented although tables to summarise the context of the paragraph would be of advantage. Moreover advantages and drawbacks should be more highlighted. The paper presented is a review of the mechanotransductive key players in the context of skin inflammatory diseases (i.e. psoriasis) and skin fibrosis as a result of chronic inflammation. Uncovering the functional role of mechanical cues and mechanotransduction in the pathogenesis of cutaneous inflammation is of fundamental importance. The review appears not too in depth of the works they cover.

We thank the reviewer for the careful analysis of the manuscript and the constructive feedback comments. Mechanisms of mechanotransduction implicated in inflammation is a very wide topic. Considering that, it is very difficult to go deep in details of each aspect. Here we tried to give a broad overview of the main components involved and to provide relevant references should the readers require more information.

1)  Comparison of all ECM components in terms of their inflammatory responses, advantages and disadvantages should be given. Studies on these components (related to stiffness of materials, cells, pathways, ECM components and everything else that the authors consider useful should be reviewed in more detail, compared and criticised. It should be taken out in (possibly) table/s. 

We now added tables describing the skin ECM components (Table 1) and integrins (Table 2) dysregulated during inflammation. Moreover, a table describing skin ECM components has been provided in ref. 21, and a table for epidermal integrins has been provided in ref. 61. Additional information on the immunomodulatory functions of the ECM can be found in a table in ref. 38.

 2) For example, Line 204: “There is ample evidence that the disruption of skin homeostasis is associated with the dysregulation of certain integrins in the epidermis (19, 61).”Which integrins? Explain.

Added to the text. More detailed information can be found in the given references.

3) On page 5 line 229 claim: “Migration of immune cells to the sites of inflammation is also dependent on the altered ECM and integrin expression in the skin (71, 72)”. Could the authors explain what exactly is altered?  

The sentence is clarified.

4) It is worth adding a short introduction/conclusion at the end or beginning of each chapter describing why this kind of components is promising to the study carried out. As well as disadvantages and limitations of works carried out.  

5) More criticism in the conclusions about all the work carried out so far and what message the authors want to send would be helpful. 

 So, the work should be systematized, which will significantly improve the level of readers perception of this review.

We believe that the manuscript has been improved and thank the reviewer again for the helpful comments.